# National Near Real-Time Vaccine Effectiveness Against COVID-19 Severe Outcomes Using the Screening Method Among Older Adults Aged ≥50 Years in Canada

**DOI:** 10.3390/vaccines14010026

**Published:** 2025-12-24

**Authors:** Robert MacTavish, Andreea Slatculescu, Dylan Ermacora, Katarina Vukovojac, Tanner Noth, Natalie Ward, Kathleen Laskoski, Daniela Fleming, Baanu Manoharan, Julie Laroche, Aissatou Fall

**Affiliations:** Vaccine Effectiveness Surveillance Program (VESPa), Vaccine Coverage and Effectiveness Surveillance Division, Centre for Immunization Surveillance and Programs, Infectious Diseases and Vaccination Programs Branch, Public Health Agency of Canada, 785 Carling Ave, Ottawa, ON K1A 0K9, Canada; robert.mactavish@phac-aspc.gc.ca (R.M.); andreea.slatculescu@hc-sc.gc.ca (A.S.); dylan.ermacora@phac-aspc.gc.ca (D.E.); katarina.vukovojac@phac-aspc.gc.ca (K.V.); tanner.noth@phac-aspc.gc.ca (T.N.); natalie.ward@phac-aspc.gc.ca (N.W.); kathleen.laskoski@phac-aspc.gc.ca (K.L.); daniela.fleming@phac-aspc.gc.ca (D.F.); baanu.manoharan@phac-aspc.gc.ca (B.M.); julie.a.laroche@phac-aspc.gc.ca (J.L.)

**Keywords:** COVID-19 vaccine effectiveness, screening method, SARS-CoV-2, severe COVID-19 outcomes, surveillance, older adults, Canada

## Abstract

**Background/Objectives:** It is critical to monitor real-world COVID-19 vaccine effectiveness (VE) in older adults, as they have been identified as a priority group for vaccination. This is the first study that aims to estimate national absolute vaccine effectiveness (aVE) against severe COVID-19 outcomes among Canadian older adults aged ≥50 years. **Methods:** The screening method (SM) was implemented using standard and spline-based logistic regression models to estimate aVE and 95% confidence intervals (CIs) by outcome, age group, vaccination status, time since last dose, vaccine schedules, and variant of concern (VOC) period. **Results:** From 1 August 2021 to 30 November 2023, there were 103,822 severe COVID-19 cases, of which 72.9% were hospitalized, 8.2% were admitted to ICU, and 18.9% had died. A total of 23.1% of these cases were unvaccinated against COVID-19, 21.9% completed a primary series only, and 55.0% received at least one additional/booster dose. National aVE against severe COVID-19 outcomes remained moderate to high during Delta and original Omicron VOC predominance periods. Monthly age-specific aVE of at least two additional/booster doses remained stable during recombinant XBB.1.5/EG.5 VOC predominance, ranging from 61.0% (95% CI: 51.9–68.4%) to 69.8% (95% CI: 67.5–72.0%) against hospitalization, and 71.0% (95% CI: 62.8–77.4%) to 77.2% (95% CI: 74.2–79.9%) against ICU admission/death. Adjusted aVE was higher for last booster doses received within the past six months and with heterologous mRNA vaccine schedules. **Conclusions:** The SM is a useful method to estimate aVE in near real-time, enabling the assessment of temporal changes in aVE, guiding vaccine policy, and building vaccine confidence among populations at higher risk of severe outcomes.

## 1. Introduction

The emergence of severe acute respiratory syndrome coronavirus 2 (SARS-CoV-2) has resulted in 4.8 million confirmed coronavirus disease (COVID-19) cases and 56,937 deaths in Canada as of 2 December 2023 [1]. COVID-19 vaccination campaigns began on 14 December 2020, initially prioritizing vulnerable populations, and have been the principal public health prevention measure to reduce the risk of severe COVID-19 health outcomes and SARS-CoV-2 transmission [2]. As of 3 December 2023, 87.3% of Canada’s total population aged ≥5 years has been vaccinated with at least one dose and 53.5% has received a primary series with at least one additional/booster dose of a Health Canada-approved COVID-19 vaccine, including the ancestral Pfizer-BioNTech/Comirnaty, Moderna Spikevax, AstraZeneca Vaxzevria, Janssen Jcovden (Johnson & Johnson), and Novavax Nuvaxovid vaccines [2]. The majority of vaccinated people received a COVID-19 ancestral monovalent or bivalent messenger ribonucleic acid (mRNA) vaccine as their last dose [2]. Epidemiologic surveillance systems have been implemented nationwide to collect data on COVID-19 cases, vaccination history, and vaccine coverage, thereby enabling researchers and government entities to continuously monitor vaccine trends and uptake [2,3,4,5].

Emerging variants of concern (VOCs) and barriers to vaccine uptake pose health risks to vulnerable populations. It is crucial to monitor real-world COVID-19 vaccine effectiveness (VE), to signal changes in vaccines’protection levels, and to inform national public health guidelines on vaccination as new variants emerge and updated vaccines are authorized. In Canada, researchers in the provinces of Ontario [6,7,8,9,10], Quebec [11], and British Columbia [12,13] have used the test-negative case–control design (TND) or the cohort study design to estimate the absolute, relative, or incremental VE against COVID-19 outcomes in adults [6,7,8,9,10,11,12,13,14,15,16] and vulnerable populations such as healthcare workers, long-term facility residents, and people with immunocompromising conditions [7,8,12,13]. Older adults, particularly those with pre-existing health conditions, are at high risk of severe outcomes and account for the largest number of hospitalizations, intensive care unit (ICU) admissions, and deaths attributed to COVID-19 in Canada [1,4,5,17]. Although COVID-19 VE has been estimated in Canadian provinces, no nationwide VE evaluation has been conducted since the roll-out of the COVID-19 vaccines at the end of 2020, and pooled VE evidence for recent VOC periods across provinces remains limited [17].

An alternative method to estimate VE is the screening method (SM), which employs a “case–cohort” or a “case-population” design that compares the proportion of cases who are vaccinated (PCV) with the proportion of persons vaccinated (PPV) in the population from which cases are drawn [18,19]. Previous studies have demonstrated the applicability of the SM for near real-time estimation of VE in pandemic settings. The SM has been used to estimate VE against influenza in France in seasonal and epidemic settings [20,21,22], and has more recently been used to estimate VE against symptomatic mpox infection in England [23]. The SM has also been adapted to estimate COVID-19 VE in other countries such as Hungary [24], the United States [25,26], Spain [27], Germany [28], Thailand [29], Malaysia [30], Kosovo [31], and the Netherlands [32]. However, existing COVID-19 VE studies using the SM have limitations [33,34,35,36], including lacking VE estimates for different Omicron sublineages or across several VOC predominance periods [24,25,26,27,29,30,31,32]; not stratifying or adjusting for age [26,27,28,29,30,31]; not comparing VE by full vaccine product series or time since last dose [24,26,27,28,29,30,31,32]; not estimating VE for varying number of doses [24,25,26,29,30,32]; and calculating crude VE without 95% confidence intervals (95% CI) [28]. What is needed is an iteration of the SM that fulfils these analysis gaps.

In this study, we built upon previous SM research and integrated COVID-19 cases and vaccine coverage surveillance data to conduct the first national COVID-19 VE study in Canada using the SM. Our study objectives were to estimate national absolute VE (aVE) against severe COVID-19 outcomes (hospitalization, ICU admission, and death) from 1 August 2021 to 30 November 2023 by age group and VOC period in older adults aged ≥50 years; to summarize monthly national VE trends per outcome, age group, vaccine product, and series; and to assess differences in VE by time since last dose administered. This work will fill research gaps and inform vaccination policy and decision making.

## 2. Materials and Methods

The SM is a pseudo-ecologic design in which two data points are needed to calculate VE: the proportion of cases vaccinated (PCV) and the proportion of persons vaccinated in the same population (PPV) [18,19,20,21,33,34,35,36]. As such, it is easy to perform with stable coverage estimates and routine data collection (e.g., provided by surveillance systems at the primary care or hospital level). Moreover, the SM can provide near real-time COVID-19 VE estimates when limited resources are available to conduct other types of observational studies [33,34,35].

### 2.1. Data Sources

We leveraged data from available administrative, immunization registry, and surveillance systems encompassing COVID-19 case, vaccination coverage, and population data in Canada. We obtained COVID-19 case data from the National COVID-19 Case dataset, a case-based surveillance system that collects data on demographics, clinical status and outcomes, risk factors, vaccination, and variant lineages of COVID-19 cases in Canada. Provinces and territories (P/Ts) report case data electronically to the Public Health Agency of Canada (PHAC) at varying frequencies. Data are subsequently mapped and stored in a Postgres (PostgreSQL) database maintained by PHAC (Metabase) [1,2,3,4,5].

Additionally, we used cumulative vaccination coverage data from P/T immunization repositories through the Canadian COVID-19 Vaccination Coverage Surveillance System (CCVCSS) at PHAC. The number of people vaccinated were aggregated by P/T, week or month, age group, vaccination status, vaccine product(s) received, and time since last dose. For this analysis, 12 out of the 13 P/Ts (excluding Québec) reported case-level vaccination data for the analysis period (from 1 August 2021 to 30 November 2023) [1,2,3,4,5]. We acquired national monthly population estimates by P/T and age from the Statistics Canada Centre for Demography client services. The unvaccinated population (no COVID-19 vaccine doses received) was calculated by subtracting the number of people with at least one dose of a COVID-19 vaccine from the population estimate [1,2,3,4,5].

Finally, we retrieved from PHAC publicly available repository VOC genomic sequencing data for a percentage of all positive COVID-19 tests performed by the Canadian COVID-19 Genomics Network (CanCOGeN), the National Microbiology Laboratory in Winnipeg, and surveillance testing from P/T laboratories [37].

### 2.2. Data Curation

COVID-19 cases were excluded from the analysis if they were outside the study period of 1 August 2021 to 30 November 2023; had missing data on key variables such as vaccination history; were not yet protected following immunization (i.e., their episode date was <14 days following their first dose of a COVID-19 vaccine) or were partially vaccinated (i.e., their episode date was ≥14 days following the first dose of a two-dose series or 0 to <14 days after receiving the second dose of a two-dose series); had missing or incorrect vaccine administration dates; or were aged <50 years (Appendix A). COVID-19 case data were stratified by outcome (hospitalization, ICU admission, and/or death), vaccination status (unvaccinated, primary series completed, primary series with one additional/booster dose received, primary series with two or more additional/booster doses received), age group (50–59 years, 60–69 years, 70–79 years, ≥80 years), and P/T and region (Central: Ontario, Quebec; Eastern: New Brunswick, Newfoundland and Labrador, Nova Scotia, Prince Edward Island; Northern: Northwest Territories, Nunavut, Yukon; Western: Manitoba, Saskatchewan, Alberta, British Columbia). Case data were also grouped by vaccine product series and time since last dose (14 days to 6 months, over 6 months) for cases reported from 1 November 2022 onwards.

Vaccination coverage data were similarly excluded from the analysis if key variables that we required for our intended population were missing. Vaccination coverage was estimated at the midpoint of each month through linear interpolation and extrapolation by age group and P/T. These data were merged with mid-month population estimates from Statistics Canada to calculate the total population by age group. The unvaccinated population was estimated per month, age group, and P/T by subtracting the population with at least one vaccine dose from the overall population. Vaccination coverage was capped at 95% to account for data limitations causing potential overestimation of vaccination coverage, such as population estimates that excluded part-time residents and coverage reporting whose quality varied between P/Ts [38]. Prior to merging with COVID-19 case data, cumulative vaccination coverage was lagged by one month to account for the time it takes for vaccine-induced immunity to develop and for the delay between COVID-19 illness onset and severe outcome presentation [21].

### 2.3. SARS-CoV-2 Variant Predominance

VOC predominance periods were calculated based on the timeframe in which ≥50% of weekly sequenced tests were due to a specified variant. To maintain continuous dates for aVE estimation, time periods during which a specific variant did not reach the 50% threshold, representing a brief transition from the deceleration of one variant and the acceleration of the subsequent variant, were split evenly between the two circulating variants. We then rounded VOC predominance dates to the nearest month to match our vaccine coverage and COVID-19 case data that were aggregated by month. Severe COVID-19 cases were reported during the predominance of Delta (1 August 2021 to 30 November 2021), Omicron B.1.1.529/BA.1/BA.2 (1 December 2021 to 30 June 2022), Omicron BA.4/BA.5/BQ (1 July 2022 to 31 January 2023), and recombinant XBB.1.5/EG.5 (1 February 2023 to 30 November 2023) VOCs.

### 2.4. Statistical Analysis

The PCV was calculated by dividing the number of cases with a specific vaccination status (excluding all other cases) by the sum of all cases who were unvaccinated and who had the specified vaccination status. Likewise, PPV was calculated by dividing the number of people in the source population with a specific vaccination status by the total population who were unvaccinated and who had the specified vaccination status. PCV and PPV were calculated by unit of time (e.g., month or VOC period), outcome, age group, and additionally, by vaccine product series and time since last dose in further sub-analyses.

To calculate 95% confidence intervals for the aVE estimates (by comparing those who received a primary series with or without an additional/booster dose with those who were unvaccinated), we first used the Farrington method [19]. This method models a standard logistic regression with PCV as the dependent variable and the logit of the matched PPV as an offset within each stratum of the variables of interest (i.e., age group, month or VOC period, time since last dose, and vaccine product).

For the monthly analyses, and to better account for months with low sample size and to adjust for confounding variables, we ran a spline-based logistic regression models (with thin plate regression splines) as proposed by Horváth et al. [24], based on a penalized least squares method approach to smooth aVE estimates over time [39]. In these models, age group was included as a covariate; vaccine product and time since last dose were also covariates. Interaction terms were included in the spline-based logistic regression approach for all available covariates, allowing different monthly trends for each unique set of covariates. These models were run for each COVID-19 outcome (hospitalization and ICU admission/death) and vaccine status.

Monthly aVE estimates were calculated only when each stratum had an adequate sample size. The minimum sample size was determined based on a conservative PPV estimate of 80%; a desired confidence interval precision of ±10% from the predicted aVE values; and predicted aVE rates of approximately 90% for severe outcomes such as ICU admission/death and 80% against hospitalization [19,21]. To produce aVE estimates, each stratum had a minimum sample size of at least 93 cases for aVE against hospitalization and at least 31 cases for aVE against ICU admission/death. Cases aged 50–59 years were excluded from the monthly aVE estimation of two or more additional/booster doses due to low case counts during Omicron BA.4/BA.5/BQ and recombinant XBB.1.5/EG.5 predominance. aVE estimations can also be influenced by rapid changes in vaccination coverage; therefore, aVE estimates for one additional/booster dose and at least two additional/booster doses were estimated one month after each P/T began reporting both case and coverage data, allowing time for coverage estimates to stabilize [21]. All data cleaning and SM analyses were conducted using RStudio 2023.12.0-369.

### 2.5. Screening Method Validation

Horváth et al. previously validated the spline-based logistic regression model using simulated epidemic data and concluded that the model accurately estimated aVE over time [24]. However, smoothing did not function as well in instances where there were low sample sizes and when there were abrupt changes in data [24]. To validate our results, we extracted aVE estimates from Canadian TND studies by VOC period, vaccine status, age group, and COVID-19 outcome, and compared them with the aVE estimates that we calculated using our spline-based SM.

## 3. Results

### 3.1. Severe COVID-19 Cases and Vaccination Status in Older Adults Aged ≥50 Years

From 1 August 2021 to 30 November 2023, across the 12 P/Ts included in our analysis, there were 103,822 severe COVID-19 cases among people aged ≥50 years who met our inclusion criteria (Appendix A).

Overall, 72.9% (75,709/103,822) of severe COVID-19 cases were hospitalized, 8.2% (8452/103,822) were admitted to the ICU, and 18.9% (19, 661/103,822) had died (Table 1). The largest number of severe COVID-19 cases were among older adults aged ≥ 80 years (45.0%; n = 46,721), followed by those aged 70–79 years (26.9%; n = 27,979), 60–69 years (18.0%; n = 18,653), and 50–59 years (10.1%; n = 10,469) (Table 1, Appendix A). Overall, people aged ≥50 years accounted for 83.4% (103,882/124,513) of all severe COVID-19 cases during the same study period.

Altogether, 10.1%, 41.8%, 31.7%, and 16.4% of severe COVID-19 cases were reported during the predominance of Delta, Omicron B.1.1.529/BA.1/BA.2, Omicron BA.4/BA.5/BQ, and recombinant XBB.1.5/EG.5 VOCs, respectively (Table 1).

A total of 23.1% (24,023/103,822) of these cases were unvaccinated against COVID-19, 21.9% (22,686/103,822) completed a primary series only, and 55.0% (57,113/103,822) received at least a primary series with one additional/booster dose (Table 1). Most vaccinated cases received any ancestral monovalent (MV) or bivalent (BV) homologous mRNA schedule for their primary series with or without an additional/booster dose (70.9%; 56,560/79,779), followed by any heterologous or mixed mRNA schedule (25.3%; 20,200/79,779), and other schedules (3.8%; 3019/79,779) (Appendix A). The most common products given as a last dose among those who recevied a primary series with or without additional/booster doses (n = 79,779) were any MV or BV Pfizer-BioNTech/Comirnaty (52.4%) vaccines, followed by any MV or BV Moderna Spikevax (32.6%) vaccines, and other vaccines (15.0%) (Appendix A).

### 3.2. National Absolute COVID-19 Vaccine Effectiveness

During Delta VOC predominance, a completed COVID-19 primary series provided high aVE across all months, age groups, and outcomes. Monthly aVE remained above 75% against hospitalization and above 85% against ICU admission/death across all age groups (Table 2, Figure 1).

The emergence of the original Omicron VOC and its subvariants resulted in a decrease in aVE against all outcomes; however, COVID-19 vaccines remained effective at preventing hospitalization and ICU admission/death. COVID-19 vaccine additional/booster doses were rolled out during Omicron subvariant B.1.1.529/BA.1/BA.2 VOC predominance. The overall aVE of a primary series completed with at least one additional/booster against hospitalization during Omicron B.1.1.529/BA.1/BA.2 VOC predominance ranged from 66.8% (95% CI: 64.7–68.8%) in those aged ≥ 80 years to 78.1% (95% CI: 76.3–79.8%) in those 60–69 years. The aVE of a primary series completed with at least one additional/booster dose against ICU admission/death ranged from 72.2% (95% CI: 69.4–74.8%) in those aged ≥ 80 years to 83.7% (95% CI: 80.1–86.7%) in those 50–59 years (Table 2). Furthermore, the monthly aVE of a primary series completed with at least one additional/booster dose decreased from February 2022 to June 2022 (Figure 1). By June 2022, aVE reached a minimum for hospitalization, ranging from 48.3% (95% CI: 36.5–57.9%) in adults aged 50–59 years to 57.0% (95% CI: 52.6–61.0%) in adults ≥ 80 years. A simlar pattern was observed for ICU admission/death, with aVE ranging from 51.6% (95% CI: 39.4–61.3%) in adults aged 70–79 years to 61.7% (95% CI: 50.8–70.1%) in adults 60–69 years (Figure 1).

During Omicron subvariant BA.4/BA.5/BQ predominance and following the National Advisory Committee on Immunization’s (NACI’s) recommendation of second additional/booster doses, the overall aVE for a primary series completed with two or more additional/booster doses was moderate to high during this vaccine rollout phase, with aVE against hospitalization ranging from 53.7% (95% CI: 43.0–62.3%) in adults aged 50–59 years to 75.0% (95% CI: 72.5–77.3%) in adults 70–79 years. aVE against ICU admission/death ranged from 47.8% (95% CI: 20.5–65.7%) in adults aged 50–59 years to 73.6% (95% CI: 66.4–79.2%) in adults 60–69 years (Table 2). The monthly aVE fluctuated slightly but remained stable at > 55% across all age groups and outcomes (Figure 1).

During Omicron recombinant XBB.1.5/EG.5 predominance, the overall aVE against hospitalization ranged from 61.0% (95% CI: 51.9–68.4%) among those aged 50–59 years to 69.8% (95% CI: 67.5–72.0%) in those ≥ 80 years. aVE against ICU admission/death ranged from 71.0% (95% CI: 62.8–77.4%) among those aged 60–69 years to 77.2% (95% CI: 74.2–79.9%) in those ≥ 80 years (Table 2). The monthly aVE against hospitalization fluctuated for all age groups except for the 50–59-year age group, which had inestimable aVE due to low monthly sample sizes. Despite these fluctuations, aVE remained above 55% and increased gradually in the final months of the analysis, coinciding with the roll out of the XBB.1.5 monovalent vaccines (Figure 1). aVE against ICU admission/death remained above 60% for these age groups and increased during this period, where estimates were available (Figure 1).

Our spline-based logistic regression models had increased precision and smoothed aVE estimates when compared in a sensitivity analysis with the Farrington standard logistic regression models, although similar aVE trends were observed using both methodologies. For monthly aVE against both hospitalization and ICU admission/death, the model without splines also had the highest aVE during Delta predominance, decreasing aVE during Omicron B.1.1.529/BA.1/BA.2 predominance, fluctuating aVE during Omicron BA.4/BA.5/BQ and recombinant XBB.1.5/EG.5 predominance, and general trends of increasing aVE by the end of 2023.

### 3.3. Product-Specific Absolute Vaccine Effectiveness

A total of 30,875 cases were included in our analysis of product-specific aVE against severe outcomes. Specifically, we estimated the effectiveness of primary series products during Delta predominance and additional/booster dose products during Omicron B.1.1.529/BA.1/BA.2 predominance. During Delta predominance, a homologous primary series of AstraZeneca/COVISHIELD had lower aVE than a homologous primary series of Moderna Spikevax or Pfizer-BioNTech/Comirnaty against both hospitalization and ICU admission/death (Figure 2). However, during Omicron B.1.1.529/BA.1/BA.2 predominance, product-specific aVE varied by outcome. aVE against hospitalization was higher for a heterologous or mixed series of three or more mRNA products than for a homologous series of three or more doses of either Pfizer-BioNTech/Comirnaty or Moderna Spikevax vaccines (Figure 3 and Appendix A). When looking at homologous series, Moderna Spikevax had higher aVE against hospitalization than Pfizer-BioNTech/Comirnaty, with a significant difference in protection among those aged ≥ 80 years (Figure 3 and Appendix A). On the other hand, aVE against ICU admission/death was similar for a heterologous mRNA and homologous Pfizer-BioNTech/Comirnaty series across all age groups; however, aVE was significantly lower for a homologous Moderna Spikevax series among those aged 70–79 years and ≥80 years. Monthly aVE against all outcomes during the predominance of Omicron subvariants from February 2022 to August 2022 declined more noticeably for a homologous primary series with additional/booster doses of either Pfizer-BioNTech/Comirnaty or Moderna Spikevax compared with a heterologous mRNA primary series followed by at least one booster/additional dose of either mRNA product (Appendix A).

### 3.4. Absolute Vaccine Effectiveness by Time Since Last Dose

aVE against hospitalization and ICU admission/death was generally higher for adults aged ≥ 60 years who received an additional/booster dose within the past 6 months compared with those who received their last additional/booster more than 6 months prior (n = 12,914) (Figure 4). For older adults who received two or more additional/booster doses, monthly aVE against hospitalization was consistently significantly higher for last doses received within the past 6 months, with estimates ranging from 70.2% (95% CI: 67.5–72.7%) to 84.6% (95% CI: 82.3–86.6%). Protection was lower when last doses were received more than 6 months prior, with aVE ranging from 54.6% (95% CI: 49.1–59.5%) to 71.7% (95% CI: 68.6–74.5%) (Figure 4). The monthly aVE against ICU admission/death was similarly higher for those who received an additional/booster dose more recently, with protection remaining significantly higher over time except for estimates observed in October to November 2023 (Figure 4). Specifically, aVE ranged from 68.6% (95% CI: 62.1–74.1) to 85.0% (95% CI: 80.2–88.6) when last doses were received within the past 6 months, whereas aVE estimates ranged from 56.3% (95% CI: 47.7–63.5%) to 79.8% (95% CI: 76.0–83.1%) when last doses were received more than 6 months prior.

## 4. Discussion

In this national analysis using Canada-wide disease surveillance and vaccination registry data, we found that COVID-19 vaccines were protective against hospitalization and severe outcomes, including ICU admission and death, in older adults aged ≥50 years. For all age groups and outcomes assessed, aVE was highest during Delta predominance (range: 84–98%) and decreased for subsequent Omicron lineages (B.1.1.529/BA.1/BA.2 range: 67–84%; BA.4/BA.5/BQ range: 47–75%; XBB.1.5/EG.5 range: 61–77%). This trend can be attributed to the increased transmissibility of novel SARS-CoV-2 variants and their ability to evade neutralizing antibodies [40], as well as other factors such as waning vaccine protection over time [10,41]. However, our monthly aVE estimates remained stable above 55% with the uptake of additional/booster doses and variant-adapted vaccines, except for those aged 50–59 years, where the small number of severe cases led to less precise estimates. This is the first Canadian study to report aVE at a national-level, providing evidence that additional/booster dose vaccination offers sustained protection against severe COVID-19 outcomes in older adults.

The aVE trends observed in our study were comparable with TND studies conducted in Ontario and Quebec (Appendix A), as well as other international studies and pooled analyses [42,43,44,45]. TND studies are recommended for estimating post-market VE, and several have been conducted in select Canadian provinces. In Ontario, two studies by Chung et al. and Nasreen et al. estimated that aVE against hospitalization or death for the Delta variant was between 95 and 97% in Ontario residents aged ≥ 60 years [6,9], which is consistent with the estimates we obtained using the SM. For the Omicron variant, aVE estimates from TND studies were more variable, depending on the total number of additional/booster doses received and the amount of time since the last dose [10,14,15,16]. Immunogenicity studies have concluded that neutralizing antibody titres increase significantly following the administration of an additional/booster dose but subsequently decrease month-to-month [46]. Carazo et al. reported aVE ranges of 66–93% (BA.1/BA.2) and 56–82% (BA.4/BA.5) for one additional/booster dose, and 82–96% (BA.1/BA.2) and 57–80% (BA.4/BA.5) for two additional/booster doses against hospitalization in Quebec residents aged ≥60 years [16]. However, studies by Lee et al. and Grewal et al. pooled hospitalization and death outcomes and reported higher aVE ranges of 87–98% for one additional/booster dose and 92–97% for two additional/booster doses during BA.1/BA.2 predominance [10,14]. aVE estimates dropped as Omicron lineages continued to diversify, ranging from 78 to 95% (BA.4/BA.5) and 47–76% (BQ/XBB.1.5) for two additional/booster doses [10,14]. We hypothesize that the higher aVE estimates in these studies compared with our results may be due to the analysis of a pooled outcome, since vaccines offer the greatest protection against death and severe illness compared to hospitalization for milder illness [44,47]. Additionally, these results were obtained from one province and were higher than TND aVE results among older adults living in Quebec [16], and so they may not be representative of other provincial or territorial populations. Furthermore, using the SM, we are unable to adjust for medical comorbidities or immune compromising conditions as these data are not collected in national surveillance systems. Therefore, our aVE estimates resemble those reported against hospitalization or death for long-term care residents, who are generally considered to be a more vulnerable population, with estimates ranging from 77 to 81% for one additional/booster dose and 83–88% for two additional/booster doses during BA.1/BA.2 circulation [8].

Similarly, international studies reported comparable aVE estimates across Delta and Omicron lineage predominance periods [42,43,44,45]. Our results were similar to aVE results in multi-country pooled TND studies by the European Centre for Disease Prevention and Control (ECDC). These studies estimated that the aVE of a two-dose series against hospitalization during Delta predominance was 90% [42], and that the aVE of a primary series with one additional/booster dose during Omicron BA.1/BA.2 predominance was 63% for those aged 60–79 years and 59% for those ≥80 years [45]. Several international studies also used the SM to estimate aVE [24,25,26,27,28,29,30,31,32,48]. In Hungary, Horváth et al. introduced the SM spline-based logistic regression models to calculate aVE against infection during Delta predominance and concluded that their results were comparable to Hungarian retrospective cohort studies [24]. Their results also showed evidence of waning aVE; however, the Horváth et al. study was limited to the Delta variant and laboratory-confirmed infections and is not directly comparable to our aVE estimates against severe outcomes across multiple VOC predominance periods. More recently, another study conducted in the Netherlands by van Werkhoven et al. used the SM to estimate VE in adults aged ≥ 60 years who received the Pfizer-BioNTech/Comirnaty monovalent XBB.1.5 vaccine following at least one prior dose compared with those who did not receive an XBB.1.5 booster. They calculated a relative VE (rVE) estimate of 70.4% (95% CI: 66.6–74.3%) against hospitalization and 73.3% (95% CI: 42.2–87.6%) against ICU admission [30]. In our findings, the aVE against severe outcomes was similar to these estimates produced by van Werkhoven et al. Further, aVE increased in those aged ≥ 60 years during the last three months of our analysis, coinciding with the authorization of the updated Pfizer-BioNTech/Comirnaty and ModernaSpikevax XBB.1.5 vaccines in September 2023 [49].

Our study also used the SM to estimate the aVE of homologous and heterologous vaccine series to determine if specific vaccine products provide more protection in older Canadians against different VOCs. We found that during Delta predominance, the aVE of a two-dose homologous primary series was higher for any mRNA vaccine compared with AstraZeneca/COVIDSHIELD. This finding is not unexpected based on previous reports showing that mRNA vaccines produce higher levels of neutralizing antibodies against COVID-19 in older adults and are associated with higher aVE against infection and severe outcomes [43,50]. During Omicron B.1.1.529/BA.1/BA.2 predominance, we focused our analysis on mRNA vaccine products because over 99% of Canadians aged ≥50 years received an mRNA booster, as per NACI recommendations. We found that a three-dose heterologous mRNA vaccine series provided more protection than a three-dose homologous mRNA vaccine series, aligning with prior research indicating that heterologous or mixed mRNA series are more immunogenic than their homologous counterparts [51]. In our analysis, a three-dose homologous Pfizer-BioNTech/Comirnaty series was associated with higher aVE than Moderna Spikevax against ICU admission and death for older adults aged 70–79 years or ≥80 years. In contrast, Breznik et al. found that three doses of Moderna Spikevax provided 47% (95% CI: 10–69%) more protection against Omicron infection than Pfizer-BioNTech/Comirnaty in long-term care residents, which may be due to a higher mRNA dose in the Moderna Spikevax formulation and/or differences in humoral and T cell responses in this population [52]. The higher aVE against hospitalization for Moderna Spikevax in adults aged ≥ 80 years that we observed is consistent with the Breznik et al. study conducted in Ontario. However, we did not observe this trend for all age groups. Additionally, the low proportion of the population receiving a homologous Moderna Spikevax series compared with a homologous Pfizer-BioNTech/Comirnaty or a heterologous mRNA series may have impacted our ability to detect differences in product-specific aVE. It is worth noting that most of the population with a heterologous mRNA series, and thus generally higher aVE against all severe outcomes, received Moderna Spikevax as an additional/booster dose rather than Pfizer-BioNTech/Comirnaty. Due to the inability to adjust for certain confounding variables, such as presence of comorbidities and time since last dose for vaccine product analyses, it is also possible the SM does not perform as well when it comes to detecting more subtle differences in aVE. We were also unable to estimate product-specific aVE for subsequent VOC periods beyond August 2022 because of changes implemented in provincial and territorial reporting of vaccine doses administered.

Lastly, our study also assessed aVE by time since last vaccine dose received. For adults who received two or more additional/booster doses, aVE against all outcomes assessed was consistently higher among those who received an additional/booster dose within the past 6 months. Our national results are consistent with literature indicating that aVE against COVID-19 infection and severe outcomes wanes over time but is restored following administration of additional/booster doses [6,10,14,43,44,53,54]. In Ontario adults aged ≥50 years without a prior COVID-19 infection, Lee et al. estimated that during Omicron BQ/XBB.1.5 predominance, the aVE of a second additional/booster dose administered in the past 3–6 months ranged from 62 to 71%, whereas the aVE of a second additional/booster ranged from 47 to 56% if received between 9 and 15 months prior [10]. Additionally, one meta-analysis by Jacobsen et al. reported that index strain neutralizing antibody titres decreased 5.9-fold (95% CI: 3.8–9.0) post booster between the first and sixth month following vaccine administration [46]. Together, these results support the recommendation for older adults to receive additional booster doses 6 months after their last dose or after a confirmed infection to maintain protection against COVID-19 over time [55].

This study is subject to limitations related to the reporting and collection of data in national surveillance systems, which may have data quality issues such as misclassification of outcomes and vaccination status, inconsistent reporting across jurisdictions, missing data on key demographic variables and/or vaccination data, and a lack of individual-level health data [24,27,33]. Due to this lack of data, our analyses could not be adjusted for known confounding factors such as prior infection, comorbidities in case patients, or community transmission rates [24,25,26,28,30,31,32,33]. However, we stratified analyses by 10-year age groups, outcomes, and VOC periods to indirectly adjust for age-related comorbidities and viral transmissibility linked with novel SARS-CoV-2 variants. We also used spline-based logistic regression models to correct time-dependent effects that introduce additional noise in the aVE estimates. Despite the data limitations regarding confounding adjustment, the similarity of our aVE estimates with TND studies and the consistency of trends observed suggest that the SM can still serve as a tool for the rapid estimation of aVE on a larger scale using less resource-intensive data that is captured by public health surveillance systems.

Another limitation is that the SM is only suitable when vaccine coverage is stable during the study period and an adequate proportion of the population is unvaccinated [32,33]. These assumptions are not always met, and for that reason, caution should be used when interpreting aVE estimates around specific events, such as the accelerated rollout of updated vaccines. In this instance, coverage is expected to change rapidly, resulting in inaccurate PPV measures and an aVE that is biased downwards. To limit these potential biases, we only included P/Ts in our analysis one month after they started providing case and coverage data by age group and vaccine status. We also capped vaccination in the population at 95% to ensure we had an appropriate denominator for aVE calculations and excluded aVE estimates for age groups with low sample sizes to avoid erroneous interpretation.

Finally, this study may also have a degree of bias due to differences in participant health-seeking behaviours. aVE estimates may be underestimated if individuals who are more likely to become infected or experience a severe outcome due to COVID-19 are also more likely to be vaccinated. Conversely, aVE may be overestimated if those at lower risk of infection, such as individuals adhering to public health measures, are more likely to be vaccinated (a phenomenon known as the “healthy-vaccinee bias”) [24,25,28,31,32,56]. We acknowledge that our study is limited by such factors, and we strongly recommend the use of the SM as a rapid detection tool alongside TND studies that are better able to assess aVE in these contexts.

Due to the quick implementation of the SM, the limited resources required, the ease with which the SM can be scaled to large populations, and the comparable aVE results to other methods, the SM is suggested as a suitable method to estimate aVE in near-real time. This allows for a more rapid assessment of the temporal changes in aVE, the results of which can guide vaccination policy and build vaccine confidence among higher risk populations.

## 5. Conclusion

Our study is the first to use the SM to estimate national monthly aVE, vaccine product-specific aVE, and time since last dose aVE against severe outcomes in Canada and builds on previous studies to expand the utility of this methodology for vaccine effectiveness studies. Understanding aVE against severe COVID-19 is important for guiding public health and policy-setting goals as severe COVID-19 has substantial repercussions on healthcare systems.

## Figures and Tables

**Figure 1 vaccines-14-00026-f001:**
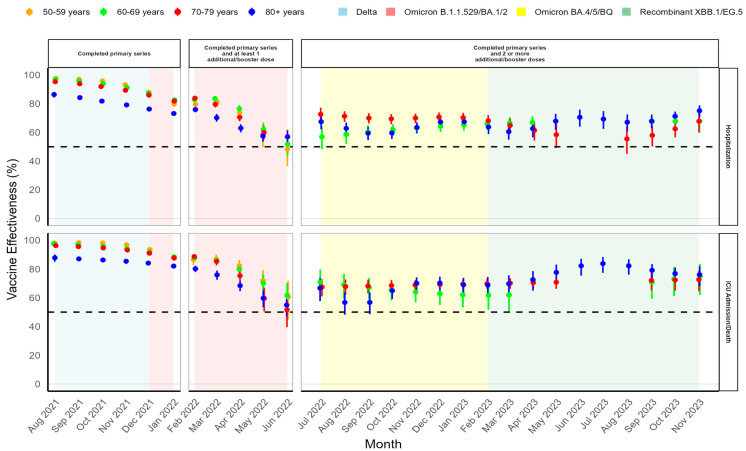
National aVE against severe COVID-19 outcomes in Canadian older adults aged ≥50 years, by month, vaccination status, and VOC, August 2021 to November 2023, using the SM spline-based model approach.

**Figure 2 vaccines-14-00026-f002:**
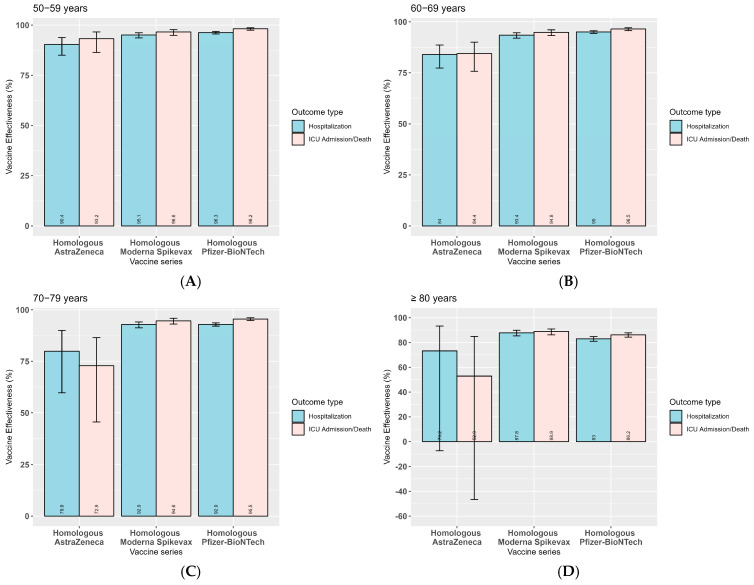
National aVE against hospitalization and ICU admission/death in older adults in Canada aged ≥50 years, by primary series during Delta predominance, August 2021 to November 2021. (**A**) 50–59 years, (**B**) 60–69 years, (**C**) 70–79 years, and (**D**) ≥80 years.

**Figure 3 vaccines-14-00026-f003:**
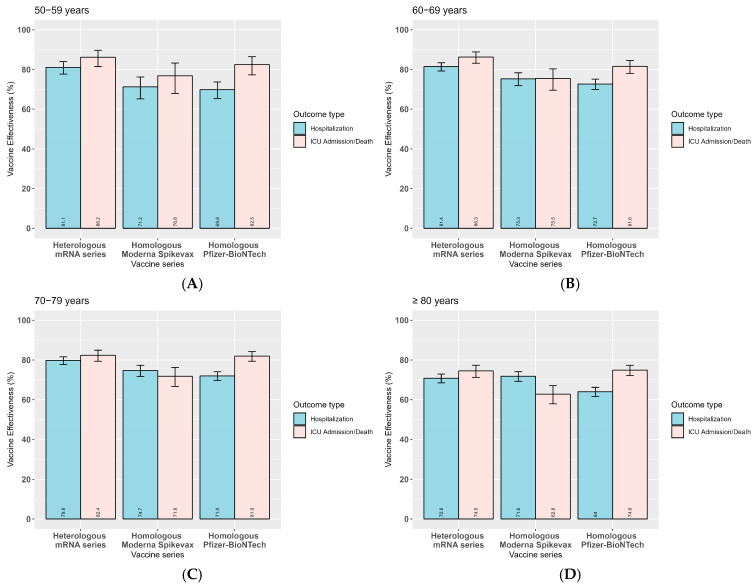
National aVE against hospitalization and ICU admission/death in Canadian older adults aged ≥50 years with a primary series and at least one additional/booster dose, by vaccine series during Omicron B.1.1.529/BA.1/BA.2 predominance, February 2022 to August 2022. (**A**) 50–59 years, (**B**) 60–69 years, (**C**) 70–79 years, and (**D**) ≥80 years.

**Figure 4 vaccines-14-00026-f004:**
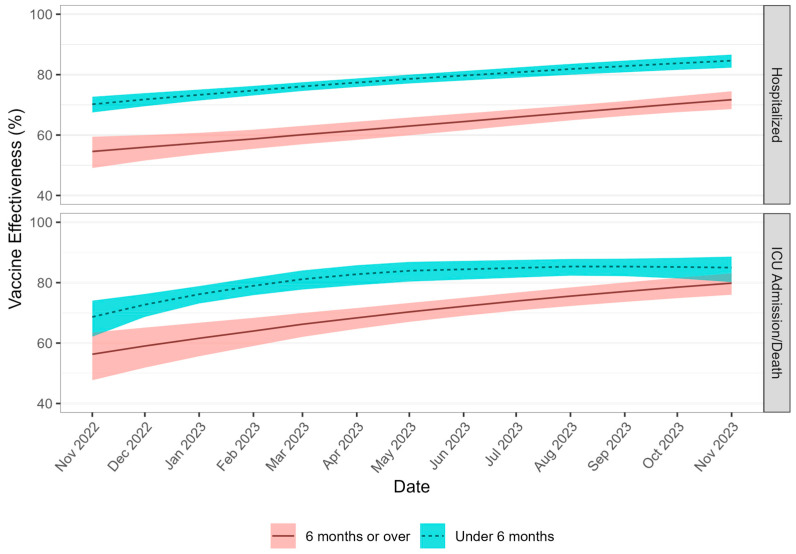
aVE against severe COVID-19 outcomes in Canadian older adults aged ≥ 60 years who completed two or more additional/booster doses, by time since last dose, November 2022 to November 2023 (n = 12,914).

**Table 1 vaccines-14-00026-t001:** Descriptive characteristics of severe COVID-19 cases by vaccination status and outcome in older adults aged ≥50 years, 1 August 2021 to 30 November 2023.

Characteristic	Overall		Unvaccinated		Primary Series Completed		Primary Series Completed with One Additional/Booster Dose		Primary Series Completed with Two or More Additional/Booster Doses	
	n	%	n	%	n	%	n	%	n	%
Totals	103,822	100	24,023	23.1	22,686	21.9	33,926	32.7	23,187	22.3
**Severe COVID-19 outcome (n = 103,822)**										
Hospitalized	75,709	72.9	16,054	66.8	16,746	73.8	25,575	75.4	17,334	74.8
Admitted to ICU	8452	8.2	2969	12.4	1947	8.6	2250	6.6	1286	5.5
Death	19,661	18.9	5000	20.8	3993	17.6	6101	18	4567	19.7
**Age group, years (n = 103,822)**										
50–59	10,469	10.1	4215	17.5	3133	13.8	2451	7.2	670	2.9
60–69	18,653	18.0	5842	24.3	5045	22.2	5350	15.8	2416	10.4
70–79	27,979	26.9	6336	26.4	6124	27.0	9253	27.3	6266	27.0
≥80	46,721	45.0	7630	31.8	8384	37.0	16,872	49.7	13,835	59.7
**Gender (n = 103,822)**										
Female	47,377	45.6	11,005	45.8	10,206	45.0	15,506	45.7	10,660	46.0
Male	56,329	54.3	12,988	54.1	12,451	54.9	18,381	54.2	12,509	54.0
Unknown/Other	116	0.1	30	0.1	29	0.1	39	0.1	18	0.0
**Province/Territory (n = 103,822)**										
Central	42,149	40.6	10,258	42.7	8738	38.5	12,000	35.4	11,153	48.1
Eastern	6718	6.5	1061	4.4	1492	6.6	2523	7.4	1642	7.1
Northern	225	0.2	60	0.2	76	0.3	59	0.2	30	0.1
Western	54,730	52.7	12,644	52.6	12,380	54.6	19,344	57	10,362	44.7
**Circulating VOC (n = 103,822)**										
Delta	10,492	10.1	7036	29.3	3351	14.8	105	0.3	–	–
Omicron B.1.1.529/BA.1/BA.2	43,411	41.8	9777	40.7	13,164	58	19,095	56.3	1375	5.9
Omicron BA.4/BA.5/BQ	32,916	31.7	4606	19.2	4411	19.4	11,205	33	12,694	54.8
Recombinant XBB.1.5/EG.5	17,003	16.4	2604	10.8	1760	7.8	3521	10.4	9118	39.3

**Abbreviations**: Values represent the total count and proportion of cases by descriptive variable and vaccination status. **ICU** = intensive care unit; **VOC** = variant of concern; **Delta** (1 August 2021 to 30 November 2021); **Omicron B.1.1.529/BA.1/BA.2** (1 December 2021 to 30 June 2022); **Omicron BA.4/BA.5/BQ** (1 July 2022 to 31 January 2023); **Recombinant XBB.1.5/EG.5** (1 February 2023 to 30 November 2023).

**Table 2 vaccines-14-00026-t002:** National aVE against severe COVID-19 outcomes by vaccination status, VOC predominance period, and age group in Canadians aged ≥50 years, August 2021 to November 2023 (n = 103,822).

	Primary Series		Primary Series with One Additional/Booster Dose		Primary Series with Two or More Additional/Booster Doses			
	Delta		Omicron B.1.1.529/BA.1/BA.2		Omicron BA.4/BA.5/BQ		Recombinant XBB.1.5/EG.5	
	aVE	*95% CI*	aVE	*95% CI*	aVE	*95% CI*	aVE	*95% CI*
**Hospitalization**								
**50–59 years**	96.4	95.9–96.9	74.9	71.8–77.6	53.7	43.0–62.3	61.0	51.9–68.4
**60–69 years**	95.2	94.7–95.7	78.1	76.3–79.8	68.9	64.5–72.6	63.7	58.3–68.3
**70–79 years**	93.3	92.6–93.9	74.3	72.3–76.1	75.0	72.5–77.3	64.3	60.6–67.7
**≥80 years**	84.5	82.8–86.1	66.8	64.7–68.8	69.6	67.4–71.7	69.8	67.5–72.0
**ICU admission/Death**								
**50–59 years**	97.9	97.4–98.4	83.7	80.1–86.7	47.8	20.5–65.7	75.8	61.1–85.0
**60–69 years**	96.5	95.9–97.0	83.5	80.9–85.7	73.6	66.4–79.2	71.0	62.8–77.4
**70–79 years**	95.5	94.8–96.1	80.4	77.8–82.6	72.3	66.9–76.9	71.9	66.1–76.6
**≥80 years**	87.1	85.4–88.5	72.2	69.4–74.8	68.9	65.2–72.2	77.2	74.2–79.9

**aVE**: absolute vaccine effectiveness. ***95% CI***: 95% confidence intervals.

## Data Availability

The case-level data are not readily available as to ensure the anonymity of cases. The aggregated results of these datasets are included in the article. Further data inquiries can be directed to the corresponding author.

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
