# Peer review of "National Near Real-Time Vaccine Effectiveness Against COVID-19 Severe Outcomes Using the Screening Method Among Older Adults Aged ≥50 Years in Canada"

_vaccines, 2025, doi:10.3390/vaccines14010026_

Round 1

Reviewer 1 Report

Comments and Suggestions for Authors

This manuscript presents an assessment of COVID-19 VE against severe outcomes among adults aged ≥50 years in Canada using the screening method. The study uses large-scale surveillance data over an extended period covering multiple variant waves and provides timely estimates that are potentially valuable for public health monitoring. I have several comments and suggestions that should be addressed prior to publication.

The authors acknowledge data limitations and the inability to adjust for key confounders. Discussion places substantial emphasis on stratification and consistency with test-negative design studies, which may give readers the impression that the resulting aVE estimates are directly comparable to causal VE estimates. It should be more explicitly stated that estimates derived from the screening method primarily serve a surveillance and monitoring purpose, and do not support a causal interpretation in the same sense as cohort or test-negative designs. In particular, age and VOC stratification do not address individual-level confounding related to prior infection, comorbidities, healthcare-seeking behavior, or exposure risk. Please clarify how biases inherent to the screening method may influence the magnitude and direction of VE estimates, especially in older age groups and during later Omicron periods.

Variant predominance is assigned based on national sequencing proportions rather than case-level linkage. This ecological classification may introduce misclassification, particularly during transition periods between variants. Please discuss the potential impact of variant misclassification on VE estimates and whether smoothing approaches may partially mask abrupt changes.

Author Response

Good morning,

Thank you for your valuable feedback. All the points mentioned have been discussed in the manuscript. The methodology we used as well as Test negative design or cohort studies are all observational studies, no causal inferences were made in our manuscript.

We also used spline-based logistic regression models to correct time-dependent effects that introduce additional noise in the aVE estimates. Despite the data limitations regarding confounding adjustment, the similarity of our aVE estimates with TND studies and the consistency of trends observed suggest that the SM can still serve as a tool for the rapid estimation of aVE on a larger scale using less resource-intensive data that is captured by public health surveillance systems.

Finally, this study may also have a degree of bias due to differences in participant health-seeking behaviours. aVE estimates may be underestimated if individuals who are more likely to become infected or experience a severe outcome due to COVID-19 are also more likely to be vaccinated. Conversely, aVE may be overestimated if those at lower risk of infection, such as individuals adhering to public health measures, are more likely to be vaccinated (a phenomenon known as the "healthy-vaccinee bias"). We acknowledge that our study is limited by such factors, and we strongly recommend the use of the SM as a rapid detection tool alongside TND studies that are better able to assess aVE in these contexts.

Horváth et al. previously validated the spline-based logistic regression model using simulated epidemic data and concluded that the model accurately estimated aVE over time. However, smoothing did not function as well in instances where there were low sample sizes and when there were abrupt changes in data.  We did not encounter the same problems due to the size of our study population.

Sincerely,

Dr Fall Aissatou

Reviewer 2 Report

Comments and Suggestions for Authors

The article submitted by MacTavish R. et al., entitled "National near real-time vaccine effectiveness against COVID-19 severe outcomes using the screening method among older adults aged ≥ 50 years in Canada" is well written and easy to understand. Objectives, methodology, results and discussion sections are well detailed and clear. Obtained results were coherent to the adopted methodology strategy.

The work is considered a huge study with the aims to estimate the national absolute vaccine effectiveness (VE) against severe COVID-19 outcomes among a population of canadian older adults aged ≥ 50 years. This objective is noble ti understand the conducting of vaccination campagnion during the CPVID-19 pandemic. Authors implemented in this work a screening method (SM) using standard and spline-based logistic regression models to estimate aVE and 95% confidence intervals (CIs) by outcome, age group, vaccination status, time since last dose, vaccine schedules, and variant of concern (VOC) period.

Intersentingly, authors highlighted by this work that The screening method used was a useful method to estimate a vaccine effectiveness in near real-time, enabling the assessment of temporal changes , guiding vaccine policy and building vaccine confidence among populations at higher risk of severe outcomes. This finding is considered as an important information for the implementation of vaccination policy for any epidemic or pandemic viral infection in any country.

The article is suitable for publication in Vaccines journal after minor revision. The proposed revision concerns:

1- Figure 3 is not clear. It should be improved.

2- Figure captions should be detailed more and information regarding the statistical results should be indicated in fig captions.

3- Figure S2 in supplemantary materials presenting the epidemic curve of severe COVID-19 cases by age group and VOC is important to understand some issues in the text. It merits to be included in the main text of the article instead to be included just in the supplemantary material.

Author Response

Good morning,

Thank you very much for taking the time to review our manuscript.

The epidemic curve has been kept in the supplementary file. Please find the updated captions for Figures 2 and 3 in track changes in the re-submitted files.

Sincerely,

Dr Fall Aissatou

Reviewer 3 Report

Comments and Suggestions for Authors

Dear Authors!

Thank you for the opportunity to review the manuscript

COVID-19 infection is a severe life-threatening disease, especially in elder population. Vaccination is one of the real tool to control the frequency and severity of the disease

The manuscript descibes the study of more than 100 000 patients is interesting and useful from the practical point of view.

The actuality is evident. The methods described in details and the results a re clear and could be reporoducible. The statistical analysis confirms the found results

The discussion contains the relevant literature and the Authors compared their results with the previously published data

The manuscript has an expanded limitation setion and the Authors disclosured all possible limitations

The Conclusion supports the study results

I have no additional comments to the manuscript

Author Response

Good morning,

Thank you very much, we appreciate the time and effort you have put into providing your valuable feedback on our manuscript.

Sincerely,

Dr Fall Aissatou

Reviewer 4 Report

Comments and Suggestions for Authors

I was invited to revise the paper entitled "National near real-time vaccine effectiveness against COVID-19 severe outcomes using the screening method among older adults aged ≥ 50 years in Canada". The aim of the study was to evaluate  national absolute VE against severe COVID-19 outcomes August 2021 to November 2023 in Canada.

It was a well conducted study with a strong methodology and a large sample study and long study period.

Authors properly described methods and results are clearly presented.

I have only some minor observations:

  • Authors should consider the possible impact of a single infection episode on the immunity status. An infection could enhance the IG titer and increase the possible protection for a longer period against worst oucomes;
  • Among discussions, Authors should compare their results with similar studies published in other IHCs;

Author Response

Good morning,

Thank you very much for taking the time to review our manuscript.

Both COVID-19 vaccination and natural infection provide strong protection against severe disease, hospitalization, and death, but they differ in mechanism, consistency, and longevity.  Several studies in Canada and elsewhere have shown that the protection conferred by hybrid immunity was more durable than that from either vaccination or prior infection alone.  As mentioned in the study, we did not have data on prior infection, although we knew that the prevalence of COVID-19 in the Omicron wave was very high in people aged ≥50 years.

We compared our results with similar studies published in other countries and in Canada.

Sincerely,

Dr Fall Aissatou